# Role of Endoscopic Ultrasound in the Diagnosis and Management of Liver Diseases

**DOI:** 10.3390/jcm14248700

**Published:** 2025-12-09

**Authors:** Mohamed Elnagar, Ira Upadhye, Madhur Varadpande, Martin W. James, Manu Nayar

**Affiliations:** 1Nottingham University Hospital NHS Trust, Nottingham NG5 1PB, UK; mohamed.elnagar@nhs.net (M.E.); martin.james4@nhs.net (M.W.J.); 2School of Medicine, Imperial College London, London SW7 2AZ, UKmadhur.varadpande20@imperial.ac.uk (M.V.); 3NIHR Nottingham Biomedical Research Centre, School of Medicine, University of Nottingham, Nottingham NG7 2RD, UK; 4HPB Unit, Freeman Hospital, Newcastle Upon Tyne and Population Health Sciences Institute, Newcastle University, Newcastle Upon Tyne NE1 7RU, UK

**Keywords:** endoscopic ultrasound, liver diseases, hepatology

## Abstract

This review explores the evolving role of endoscopic ultrasound (EUS) in the diagnosis and management of liver diseases, with a particular focus on chronic liver disease, focal hepatic lesions, portal hypertension, and post-transplant anatomy. A comprehensive literature review of PubMed, MEDLINE, and Embase studies up to August 2025 was conducted to identify the latest evidence on EUS-guided procedures, comparing them with traditional techniques. In diagnostics, EUS-guided liver biopsy provides real-time visualisation and precise tissue sampling, achieving longer specimen lengths and better patient outcomes compared to traditional percutaneous and transjugular approaches. For portal hypertension assessment, EUS-guided portal pressure gradient measurement is a promising alternative to conventional methods, with validation studies demonstrating strong correlation with hepatic venous pressure gradient measurements. In therapeutic applications, EUS facilitates precise interventions including gastric variceal treatment through combined coil and glue injection, management of visceral arterial pseudoaneurysms, selective portal vein embolisation, and targeted tumour ablation. While some applications remain in developmental stages, studies support the safety and efficacy of EUS in improving diagnostic accuracy and expanding therapeutic options for liver diseases. Ongoing technological advances in needle design, imaging capabilities, and artificial intelligence integration are expected to further enhance the utility of EUS in hepatology.

## 1. Introduction

Endoscopic ultrasound (EUS) has become a valuable tool in the diagnosis, staging and management of the spectrum of liver diseases, including chronic liver conditions such as metabolic-associated steatotic liver disease (MASLD), viral hepatitis, autoimmune hepatitis, and alcoholic liver disease. Despite the emergence of EUS as a multi-faceted tool, a single, comprehensive review that combines the latest evidence across diagnostics (biopsy, pressure measurement, lesion detection) and a wide aspects of therapeutics (variceal treatment, tumour ablation, portal vein embolization) remains an unmet need. This review fills that gap by providing a current, holistic perspective on EUS in hepatology, with a focus on its comparative advantages over conventional methods and the latest technological advances, such as contrast-enhanced EUS. While conventional imaging techniques such as ultrasound scanning (USS), computed tomography (CT) and magnetic resonance imaging (MRI) are commonly used to detect liver abnormalities, they have limitations in detecting small or complex lesions. EUS overcomes these challenges by providing real-time imaging of the liver and surrounding blood vessels. It also allows for target tissue sampling through fine-needle aspiration or biopsy, allowing for diagnostic accuracy. Beyond diagnosis, EUS is playing a growing role in treatment. It can be used for noninvasive liver fibrosis assessment, guided portal pressure measurement, cyst drainage, and even targeted ablation. Because it is minimally invasive, it has minimal adverse effects and thus improves patient care. As EUS technology advances and its application expands, an up-to-date review is needed to highlight its evolving role in the diagnosis and management of liver disease. This article is designed to be accessible to non-endoscopists and non-gastroenterologists, aiming to explore the latest developments in EUS and how it compares to traditional imaging methods.

## 2. Methods

To write this review, we looked at the latest research on how endoscopic ultrasound (EUS) is used to diagnose and treat liver diseases—especially chronic liver conditions, focal liver lesions, portal hypertension, and cases with altered anatomy like liver transplant recipients. A comprehensive literature review included searching PubMed, MEDLINE, and Embase for studies published from database inception up to August 2025. Our search strategy employed a combination of MeSH terms and keywords using the Boolean operator “AND”. Key search terms included “Endoscopic Ultrasound” AND “Liver Diseases” OR “Hepatology”, supplemented by specific procedure terms like “EUS-guided liver biopsy”, “portal pressure measurement”, “variceal treatment”, “tumour ablation,” and “portal vein embolization”. Our focus was on research that studies EUS-guided liver biopsy, pressure measurements, lesion detection, and therapeutic procedures like variceal treatment or tumour ablation. Where available, clinical guidelines and meta-analyses were included. We prioritised studies that involved adult patients and were published in English. Case reports with very small sample sizes were excluded. The findings were summarised to highlight how EUS compares with traditional techniques, what benefits it offers, and where it is still evolving, as well as emerging technologies like contrast-enhanced EUS and artificial intelligence, which may shape future practice.

## 3. Diagnostic EUS in Liver Diseases

### 3.1. EUS-Guided Liver Biopsy and Ancillary Techniques

Liver biopsy remains the gold standard for diagnosing and staging chronic liver diseases [1]. Traditionally, it has been performed using two techniques: percutaneous (PC-LB) and transjugular (TJ-LB). The percutaneous method, first described in 1883, involves percussing the liver and inserting a 16- or 18-gauge Menghini needle into the right lobe under local anaesthesia [2]. Modern versions of PC-LB use Trucut or Biopince needles, which have been developed to improve diagnostic accuracy and reduce complications [3]. TJ-LB, introduced in the late 1900s, is performed by interventional radiologists via the jugular vein. It is preferred in patients with an increased risk of bleeding, such as those with coagulopathy or ascites, as it confines bleeding to the vascular system. It also allows measurement of the hepatic venous pressure gradient (HVPG), which is a marker of portal hypertension [4]. However, both PC-LB and TJ-LB lack real-time visualisation, which limits their target sampling.

EUS-guided liver biopsy (EUS-LB) has emerged as a practical alternative, providing real-time guidance and access to both liver lobes. This technique uses a linear-array echoendoscope with a 19-gauge fine needle biopsy (FNB), and colour Doppler imaging to ensure a vessel-free trajectory for precise tissue sampling. EUS-LB is particularly beneficial in patients with ascites or challenging body habitus, demonstrating a high diagnostic yield with low complication rates [5]. Comparisons between EUS-LB and PC-LB are limited, as most of the existing research predates 2020 and relied on first-generation fine-needle aspiration needles [6,7].

The development of second-generation FNB needles has significantly improved tissue yield and diagnostic accuracy [8,9]. Advances in biopsy needle design, such as Franseen and Fork-tip 19-gauge FNB needles, have further enhanced tissue adequacy and core specimen length [10]. Furthermore, the introduction of ‘wet’ suction techniques, in which the stylet is removed, and the needle is flushed with a heparin solution, has further improved the quality of tissue samples obtained during EUS-LB [11]. A recent multicenter study compared two techniques for EUS-guided pancreatic lesional biopsy: wet suction and slow-pull. The study included 210 patients and used 22G fork-tip or Franseen needles. Wet suction gave more complete tissue samples (71.4% vs. 61.4%) and better tissue quality but also caused more blood in the samples. For liver lesions, both techniques gave similar results. These findings suggest wet suction may help get better samples, but more studies focused on the liver are needed [12].

The British Society of Gastroenterology [13] and the American Association for the Study of Liver Disease [14] define tissue adequacy as a core length greater than 20–30 mm and greater than 11 complete portal tracts (CPTs). Studies have demonstrated the advantages of EUS-LB over traditional biopsy methods. For example, Pineda et al. reported that EUS-LB yielded superior tissue samples compared to PC-LB and TJ-LB [15]. Specifically, EUS-LB obtained significantly more tissue in both liver regions, with greater total specimen length (TSL) and CPTs than PC-LB (*p* = 0.0000 and 0.0006). Additionally, EUS-LB produced longer TSL than TJ-LB (*p* = 0.01) while achieving similar CPTs. Another study found that EUS-LB achieved longer aggregate specimen lengths (36.9 mm vs. 17.7 mm for PC-LB and 13.5 mm for TJ-LB) and higher CPT counts (9.0 vs. 7.7 and 6.8, respectively) [16]. EUS-LB can be used for parenchymal liver biopsy to diagnose disease aetiology and stage liver fibrosis, as well as for targeting focal liver lesions.

EUS-LB offers practical advantages, including shorter recovery times of 90 to 120 min, compared to up to 4 h for PC-LB and approximately 140 min for TJ-LB. Additionally, EUS-LB is associated with reduced pain and higher patient satisfaction [10,17,18]. A recent randomized controlled trial by Benmassaoud et al. (2025) [19] compared EUS-guided liver biopsy with the transjugular approach. The study showed that EUS achieved longer biopsy specimens, more complete portal tracts with shorter procedure time and greater patient satisfaction [19]. Complication rates are low across all biopsy methods, with haemorrhage being the most significant [20]. However, EUS-LB minimises this risk through real-time imaging, enabling precise needle placement while avoiding vascular structures. Care must be taken to prevent accidental splenic puncture, particularly in patients with hepatic steatosis, where distinguishing between the liver and spleen can be challenging [21]. Collectively, the data suggests EUS-LB is superior in minimizing risks while enhancing tissue yield and patient experience (Table 1).

Emerging EUS-based technologies such as shear-wave elastography (EUS-SWE) and contrast-enhanced EUS (CE-EUS) are expanding the role of EUS in liver assessment. Early data suggest that EUS-SWE may help characterise focal liver lesions and estimate liver fibrosis, although its use remains largely research-focused [22,23]. CE-EUS can enhance visualisation of small left-lobe lesions and improve differentiation between benign and malignant findings, particularly when conventional imaging is limited [24]. These techniques may also be helpful in patients with altered anatomy, such as those with MASLD or post-transplant.

### 3.2. EUS-Guided Evaluation of Liver Lesions

CT and MRI are standard modalities for detecting primary liver lesions, but their sensitivity is limited for lesions smaller than 2 cm. In contrast, EUS offers a minimally invasive alternative capable of detecting lesions as small as 1 cm, with real-time imaging that facilitates precise tissue acquisition via FNA or FNB. EUS is particularly effective in visualising the left and caudate lobes, which are often challenging to assess with conventional imaging [25].

A study by Singh et al. found that EUS outperforms CT in detecting hepatic metastases (98% vs. 92%) and identifying a greater number of metastatic lesions (40 vs. 19) [26]. To differentiate benign from malignant lesions, Fujii-Lau et al. [27] proposed diagnostic criteria, where malignancy is suggested if at least three of the following features are present: (i) absence of an isoechoic or slightly hyperechoic centre, (ii) post-acoustic enhancement, (iii) distortion of adjacent structures, (iv) hypo echogenicity, and (v) lesion size of at least 10 mm.

EUS-guided FNA and FNB have demonstrated high diagnostic accuracy, with sensitivity ranging from 89% to 100% and specificity up to 100% for malignancies [20]. EUS-FNA has been shown to influence clinical management in 86% of patients with confirmed malignancies [28]. While FNA is more commonly used, FNB is preferred for core tissue sampling due to its ability to provide higher-quality specimens with fewer needle passes. Studies also suggest that FNB results in fewer inadequate samples compared to FNA [29,30]. Patients who had FNB had shorter procedure times (37.4 vs. 44.9 min, *p* < 0.001) and required fewer passes (2.9 vs. 3.8, *p* < 0.001). The cytologic diagnostic yield was also higher with FNB compared to the FNA group (98.3% vs. 90.2%, *p* = 0.003). The adverse event rates were similar (1.1% vs. 0.5%, *p* = 0.564). However, the limited availability of FNB needles and trained specialists remains a challenge [29].

### 3.3. EUS-Guided Portal Pressure Gradient Measurements

Portal hypertension is a key factor in the progression and prognosis of liver disease and the development of complications such as ascites, spontaneous bacterial peritonitis, variceal bleeding, and hepatic encephalopathy [31]. This is especially relevant today, given the increasing prevalence of MASLD. Elevated portal pressure is also associated with an increased risk of hepatic decompensation and mortality following surgical resection [32]. Measuring the portal pressure gradient provides valuable staging and prognostic information and can help guide treatment decisions, such as the use of beta-blockers or the assessment of therapeutic response in clinical trials.

Various methods have been developed to measure portal pressure. Initially, direct access to the portal vein was achieved through transhepatic puncture, umbilical vein catheterisation, or intraoperative techniques, though these invasive approaches are now rarely used. The current gold standard method involves interventional radiologists accessing the internal jugular vein to measure free and balloon-wedged hepatic venous pressures, enabling calculation of the HVPG. However, these techniques require fluoroscopy, are technically challenging, and have limited availability outside specialised centres. Furthermore, HVPG is an indirect measurement of portal pressure and may underestimate true values, particularly in pre-sinusoidal portal hypertension or MASLD [33].

EUS offers a minimally invasive alternative by utilising the proximity of the echoendoscope in the proximal stomach to the portal and hepatic veins. This allows direct measurement of portal pressure. The EUS-derived portal pressure gradient (EUS-PPG) is calculated by subtracting the mean hepatic vein pressure (HVP) from the portal vein pressure (PVP) [34].

HVP is measured by inserting a 25-gauge or 22-gauge fine-needle aspiration (FNA) needle into the left or middle hepatic vein, which is connected to a manometer with non-compressible saline-filled tubing. When hepatic vein access is challenging, the intrahepatic inferior vena cava serves as an alternative measurement site. Similarly, PVP is obtained by targeting the umbilical portion of the left portal vein or using a transhepatic approach at the liver hilum.

Clinical studies have demonstrated the feasibility and accuracy of EUS-PPG. A pilot study by Huang et al. successfully performed pressure measurements in all 28 patients without adverse events [35]. PPG ranged from 1.5 to 19 mm Hg and showed a strong link to key signs of portal hypertension, such as varices, PH gastropathy and thrombocytopenia. Zhang et al. reported a success rate of 91.7% in 11 patients, also with no complications [36]. A larger study by Choi et al., involving 83 patients, achieved a 100% technical success rate [37]. Furthermore, a prospective validation study comparing EUS-PPG with HVPG reported a strong intraclass correlation coefficient (ICC 0.82, 95% CI: 0.65–0.91), confirming its accuracy and safety [38]. This study showed EUS-PPG is safe and matches well with HVPG, but it had some limitations. It was performed at one expert centre, so the results may not apply everywhere. The number of patients was small, and it did not look at long-term outcomes. Also, the technique needs skilled operators and there is no standard way to do it yet. More studies are needed before it can be used widely. A recent bicentric European study by Vanderschueren et al. compared EUS-guided portal pressure gradient (EUS-PPG) measurement with the transjugular hepatic venous pressure gradient (HVPG) in patients with cirrhosis. The study demonstrated an excellent correlation between the two techniques (r = 0.74, *p* = 0.0001), with comparable accuracy and no adverse events, confirming EUS-PPG as a safe and reliable alternative for the assessment of portal hypertension [39]. Similarly, a randomized controlled trial by Benmassaoud et al. compared EUS-PPG with the standard transjugular approach, showing that EUS-PPG provided highly consistent and reproducible results, strongly correlating with HVPG measurements. The EUS approach achieved high technical success, shorter procedure time, and superior patient tolerability, further supporting its feasibility as a minimally invasive and efficient alternative for evaluating portal pressure [19] (Table 2).

### 3.4. EUS-Guided Portal Vein Sampling

The ability to access the portal vein directly using EUS offers additional benefits, such as sampling portal vein blood. A small pilot study on metabolite profiling demonstrated that EUS-guided sampling is safe in both cirrhotic and non-cirrhotic patients [40]. Portal vein samples were of comparable quality to those collected from the systemic circulation [40]. Future applications could include using this technique as a diagnostic or prognostic tool to assess cancer risk, disease recurrence, or metastasis development. For instance, portal vein portal venous circulating tumor cells can be identified and used for the molecular characterisation of pancreaticobiliary cancers (PBCs) and therefore be used to analyse the pathogenesis and progression of PBCs [41]. Additionally, preclinical data suggest that EUS-guided portal vein injection of chemotherapy is effective and holds promise for treating hepatic metastases [16]. Other innovative interventions involving the portal vein, such as EUS-guided portosystemic shunts and selective portal vein embolisation, have shown favourable results in preliminary animal studies [42]. However, further human studies and the development of appropriate accessories are needed to confirm their safety and effectiveness.

## 4. Therapeutic EUS in Liver Diseases

### 4.1. EUS-Guided Vascular Interventions

EUS is increasingly being used for therapeutic purposes. EUS-guided vascular intervention is a minimally invasive approach that leverages the use of high-resolution Doppler ultrasound to guide procedures. The first documented vascular intervention was by Lahoti et al. in 2000, when five patients underwent sclerotherapy for oesophageal varices [43]. In this procedure, patients required an average of 2.2 sessions to achieve complete obliteration of varices, and no bleeding or deaths occurred in these procedures. Since then, there have been significant advances in the field, with EUS-guided vascular interventions now also being used for tissue acquisition, portal pressure measurement and variceal obliteration.

Gastric varices are dilated submucosal collateral veins that occur due to portal hypertension or extrahepatic portal vein obstruction [44]. They are less common than oesophageal varices but are more concerning due to higher rates of life-threatening bleeding, with an incidence of 22.7% [45,46]. Treatment typically involves direct endoscopic injection (DEI) of cyanoacrylate glue into the varix, which polymerises upon contact with blood to promote haemostasis. However, on endoscopy, the submucosal vessel can be difficult to visualise, especially with active bleeding or clots in the stomach. An EUS-guided approach uses colour Doppler to precisely identify the target varix, allowing for injection of glue or thrombin via a 19- or 22-gauge FNA needle. This method also provides real-time feedback on variceal obliteration by showing reduced Doppler flow. A meta-analysis by Mohan et al. reported that the EUS approach was effective in treating gastric variceal bleeding [47] and significantly better than a DEI approach in achieving variceal obliteration. Moreover, EUS facilitates the introduction of haemostatic coils, originally used in IR to manage bleeding. Mohan et al. also found that using both coil and glue in an EUS approach was more effective than using either agent alone. Building on these promising results, future research is needed to determine the optimal treatment technique, the best agents to use, and the appropriate size and number of coils [48].

Visceral arterial pseudoaneurysms are abnormally dilated arteries with thin vessel walls, typically resulting from inflammation (such as acute pancreatitis) or following surgical resection or trauma. The most affected vessels in acute pancreatitis are the splenic, gastroduodenal, or pancreaticoduodenal arteries. While rare, pseudoaneurysms carry high risks of rupture and bleeding, leading to significant mortality. Small pseudoaneurysms can be challenging to visualise using traditional surgical or IR methods. An EUS-guided approach can be used to identify and treat pseudoaneurysms, often through injection of glue and coil placement in the aneurysm sac [49]. The largest study on this intervention, a case series by Maharshi et al. of eight patients, demonstrated promising potential for this technique [50]. The study used EUS-guided thrombin injection to treat pseudoaneurysms, and it was successful in all patients. After 3 months, follow-up showed that 7 patients (87.5%) had their pseudoaneurysms completely resolved. However, complications such as systemic arterial embolism or coil migration must be considered, making careful selection of injectate and coil sizing crucial. Due to the rare nature of these pseudoaneurysms and the limited number of studies, treatment should be individualised with careful consideration of surgery, IR or therapeutic EUS approaches (Table 3).

### 4.2. EUS-Guided Portal Vein Embolisation

Pre-operative portal vein embolization (PVE) induces atrophy of the treated liver lobe and compensatory hypertrophy of the non-embolised remnant liver. This process increases future liver volume, which reduces the risk of postoperative liver dysfunction, enabling more permissive curative hepatic resections. It is commonly performed in patients with hepatocellular carcinoma (HCC), intrahepatic or hilar cholangiocarcinoma or colorectal liver metastases receiving extensive liver resection [51]. Liver resection is typically performed two to six weeks after PVE to allow adequate compensatory hypertrophy of the future liver remnant. Endosonographers require meticulous knowledge of liver and portal venous system segmental anatomy before performing this procedure [42].

Currently, PVE is most performed using a percutaneous transhepatic approach by vascular interventional radiologists [51]. In a porcine model designed to study EUS-guided selective intrahepatic PVE, the portal vein was first punctured with a 19-gauge FNA needle. A microcoil was inserted, followed by cyanoacrylate (CYA) glue injected through the same FNA needle. Doppler imaging was used to assess portal blood flow. The success rates for coil and CYA delivery were 88.9% and 87.5%, respectively. In one case, the embolised coil migrated into the hepatic parenchyma, and CYA injection failed due to clogging in the FNA needle. One week later, postoperative necropsy showed total occlusion of the selected portal vein with embolus with no damage to other organs [51]. Further studies are needed to compare the EUS-guided PVE with the percutaneous approach and evaluate the long-term outcomes.

### 4.3. EUS-Guided Treatment of Liver Tumours

EUS-guided fine-needle injection (FNI) of ablative agents such as ethanol or chemotherapeutic drugs has shown potential for treating liver tumours, particularly those that are difficult to access [52]. Ethanol injection is commonly used in percutaneous therapy, and it induces tissue necrosis and vasculitis, which in turn leads to reduced recurrence rates. This method is effective in treating HCC and metastatic liver lesions, with needle sizes ranging from 18- to 25-gauge. Studies have demonstrated the efficacy of EUS-guided ethanol injection in these cases, highlighting its potential as a targeted, minimally invasive treatment option for liver tumours [52]. In addition to ethanol injection, thermal therapies like EUS-guided radiofrequency ablation (RFA), laser ablation, and photodynamic therapy are emerging as viable treatment options for liver tumours while preserving surrounding healthy tissue. Though human studies are limited, animal models show promising results. RFA works by using alternating current to generate heat, which is effective in targeting challenging lesions, particularly in obese patients or those with difficult-to-reach liver lobes. This technique is gaining recognition for its ability to treat lesions that are inaccessible with percutaneous methods [53]. Cryoablation is another promising thermal therapy which involves freezing and thawing tissue to induce cell death and has shown positive results in animal trials [54]. Laser interstitial thermal therapy uses focused laser heat to destroy tumour cells, with some studies reporting complete tumour resolution over time [55]. Photodynamic therapy, which involves the injection of photosensitising agents followed by light exposure, induces dose-dependent tissue necrosis [56]. Before EUS-guided thermal therapies and injection treatments can become mainstream clinical options, more human trials are needed to fully understand their potential benefits, risks, and optimal application in treating liver tumours.

## 5. Future Directions

While the role of EUS in managing liver diseases (Table 4) is an exciting development, several key areas need to be addressed before it becomes a routine part of clinical practice. First, larger-scale, randomized controlled trials (RCTs) are needed to demonstrate its superiority or at least non-inferiority when compared to standard interventional radiology techniques. The ramifications of current findings point toward EUS becoming the primary access route for vascular procedures in the liver. However, RCTs must confirm that EUS-guided treatments like portal vein embolisation and tumour ablation offer equivalent or better long-term oncological and functional outcomes than established percutaneous or surgical approaches. Additionally, while technical feasibility is an important consideration, it does not always translate to practical convenience, and in some cases, radiological or surgical interventions may still be preferred. Another key point to consider is determining who should perform these procedures, as there are currently no defined standards for attaining and maintaining competency. Questions remain regarding specific training programmes, competency requirements, and established standards, with competency in interventional EUS inevitably requiring training and mentorship in high-volume interventional endoscopy units. Furthermore, most available data on these procedures come from tertiary academic centers with expertise in complex EUS, which may limit their implementation in real-world or community settings. Lastly, the future will be shaped by technological advances and integration with artificial intelligence (AI). AI could play a crucial role in evaluating these lesions and, more importantly, guiding clinicians in decision-making and patient management.

## 6. Conclusions

EUS has significantly improved the diagnosis and management of liver diseases, providing high-resolution imaging, real-time guidance, and emerging therapeutic options. Compared to traditional imaging or interventional radiology methods, EUS offers superior precision, safety, and diagnostic yield, especially when dealing with small or hard-to-reach lesions. While challenges remain, ongoing advancements in EUS technology, especially those in AI and molecular diagnostics, are expected to further enhance diagnostic accuracy, providing more detailed information and expanding therapeutic options in the management and treatment of liver diseases.

## Figures and Tables

**Table 1 jcm-14-08700-t001:** Summary of Comparative Studies Evaluating EUS-Guided Liver Biopsy (EUS-LB) Outcomes.

Study	Study Design	Biopsy Method	Needle Type	Specimen Length (mm)	Complete Portal Tracts (CPTs)	Complication Rate
“Mohan et al. [5]”	Primary Study	EUS-LB	Not specified	Not specified	Not specified	2.3%
“Ali et al. [7]”	RCT	EUS-LB vs. PC-LB	Not specified	Not specified	Not specified	Comparable
“Pineda et al. [15]”	Retrospective	EUS-LB vs. PC-LB & TJ-LB	19G FNA (EUS)	36.9 vs. 17.7 vs. 13.5	9.0 vs. 7.7 vs. 6.8	Low
“Benmassaoud et al. [19]”	RCT	EUS-LB vs. TJ-LB	19G FNA (EUS)	117 vs. 29.2	29.2 vs. 11.2	6.9% (EUS) vs. 10.3% (TJ)

Abbreviations: EUS-LB, endoscopic ultrasound-guided liver biopsy; PC-LB, percutaneous liver biopsy; TJ-LB, transjugular liver biopsy; CPTs, complete portal tracts; RCT, randomised controlled trial; FNA, fine-needle aspiration.

**Table 2 jcm-14-08700-t002:** Studies on the feasibility and accuracy of EUS-derived portal pressure gradient (EUS-PPG) measurements.

Study	Sample Size	Technical Success	Correlation with HVPG	Adverse Events
“Benmassaoud et al. [19]”	58	90% (EUS) vs. 96.6% (TJ)	Strong correlation	6.9% (EUS) vs. 10.3% (TJ)
“Huang et al. [35]”	28	100%	Not reported (correlated with clinical markers)	None
“Zhang et al. [36]”	12	91.7%	r = 0.923	None
“Choi et al. [37]”	64	100%	Not reported (correlated with clinical markers)	None (Serious)
“Vanderschueren et al. [39]”	21	100% (ENCOUNTER Study)	r = 0.74 (*p* = 0.0001)	None

Abbreviations: EUS-PPG, endoscopic ultrasound-guided portal pressure gradient; HVPG, hepatic venous pressure gradient.

**Table 3 jcm-14-08700-t003:** Summary of EUS-guided vascular intervention studies.

Study	Intervention	Target	Success Rate	Complications
“Mohan et al. [47]”	EUS-guided Coil + (CYA) Glue	Gastric Varices (GV)	100% Technical Success, 90% Clinical Success	None reported
“Maharshi et al. [50]”	EUS-guided Thrombin Injection	Visceral Artery Pseudoaneurysms (VAPA)	100% Technical Success, 87.5% Obliteration Rate	None reported

Abbreviations: EUS, endoscopic ultrasound; CYA, cyanoacrylate.

**Table 4 jcm-14-08700-t004:** Overview of EUS uses in liver diseases.

Application Area	Technique	Advantages	Limitations
Liver Biopsy	EUS-guided liver biopsy (EUS-LB) [5,7,15,19]	Real-time imaging, access to both lobes, longer specimen length, fewer complications	Limited RCTs, requires advanced needle design and operator expertise
Fibrosis Assessment	EUS shear wave elastography (EUS-SWE) [22,23]	Potential for real-time fibrosis quantification	Currently limited to research; not widely validated
Contrast Imaging	Contrast-enhanced EUS (CE-EUS) [24]	Improves lesion detection and characterisation	Limited by anatomical reach; mostly left lobe
Lesion Detection and Characterization	EUS imaging + FNA/FNB [20,26,27]	Detects lesions < 1 cm, high sensitivity/specificity, especially in left/caudate lobes	Limited access to right lobe; requires trained personnel and FNB needle availability
Portal Hypertension Evaluation	EUS-guided portal pressure gradient (EUS-PPG) [19,37,39]	Direct pressure measurement, strong correlation with HVPG, safe and feasible	Needs further validation for routine use
Portal Vein Sampling	EUS-guided blood sampling from portal vein [40,41]	Safe in cirrhotic/non-cirrhotic patients, potential for cancer profiling	Early-stage technique; limited human data
Vascular Interventions	EUS-guided coil/glue injection for gastric varices [47]	Precise targeting, real-time feedback, superior to direct endoscopic injection	Risk of embolism or coil migration; requires Doppler guidance
Pseudoaneurysm Management	EUS-guided thrombin or glue injection [50]	Effective in small, hard-to-access pseudoaneurysms	Rare condition; limited studies; risk of embolism
Portal Vein Embolisation (PVE)	EUS-guided selective intrahepatic PVE [42,51]	Induces hypertrophy of liver remnant pre-resection	Mostly animal data; technical challenges with coil/glue delivery
Tumour Ablation	EUS-guided ethanol injection, RFA, laser, cryoablation, photodynamic therapy [52,53,54,56]	Targeted therapy for inaccessible lesions, minimally invasive	Mostly preclinical; human trials needed

## Data Availability

The original contributions presented in this study are included in the article. Further inquiries can be directed to the corresponding author.

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
