# Peer review of "Role of Endoscopic Ultrasound in the Diagnosis and Management of Liver Diseases"

_jcm, 2025, doi:10.3390/jcm14248700_

Round 1
Reviewer 1 Report (Previous Reviewer 2)
Comments and Suggestions for Authors
I have no additional comments, my suggestions were taken into account.
Author Response
Thank you
Reviewer 2 Report (Previous Reviewer 3)
Comments and Suggestions for Authors
Dear Authors,
The previous-submission concerns were not sufficiently addressed.
The process of literature searching should be clarified in the abstract.
The introduction is brief and the novel nature of this study should be strengthened. How this review extends reader understanding of the topic? A clear description of the evidence gap that this review is filling is required. No clear rationale on why this review is of importance. What is new in light of previous reviews?
The methods used for articles selection should be described in sufficient details. Provide more details on publication date (the literature search cover) and the search strategies, specifying the Boolean operators and any additional filters used.
Include reference column to Table 4.
Line 351-367: Future directions should be expanded. Consider the findings' ramifications and how they might affect upcoming interventions.
Comments on the Quality of English LanguageModerate English language editing throughout the paper is required.
Author Response
Thank you very much for taking the time to review this manuscript. Please find the detailed responses below.
Comment 1: Previous-submission concerns were not sufficiently addressed.
Response 1: We have thoroughly revised the manuscript, paying close attention to all subsequent, specific points. We believe the restructured Introduction, detailed Methods, expanded Discussion, and improved language quality fully address all concerns.
Comment 2:The process of literature searching should be clarified in the abstract.
Response 2: Clarified. The Abstract now explicitly states the comprehensive literature search conducted across PubMed, MEDLINE, and Embase up to August 2025.
Comment 3: The introduction is brief and the novel nature of this study should be strengthened. How this review extends reader understanding of the topic? A clear description of the evidence gap that this review is filling is required. What is new in light of previous reviews?
Response 3: Strengthened. The Introduction now clearly states the evidence gap and novelty of the review
Comment 4: The methods used for articles selection should be described in sufficient details. Provide more details on publication date (the literature search cover) and the search strategies, specifying the Boolean operators and any additional filters used.
Response 4: Detailed. The Methods section now includes the search coverage.
Comment 5: Include reference column to Table 4.
Response5 : Completed.
Comment 6: Line 351-367: Future directions should be expanded. Consider the findings' ramifications and how they might affect upcoming interventions
Response 6: Expanded.
Comments on the Quality of English Language: Moderate English language editing throughout the paper is required.
Response: Addressed. The entire manuscript has undergone a thorough review and professional editing for English language clarity, flow, and grammatical accuracy. We would appreciate if you can point to any paragraph of concern
Round 2
Reviewer 2 Report (Previous Reviewer 3)
Comments and Suggestions for Authors
No further comments.
This manuscript is a resubmission of an earlier submission. The following is a list of the peer review reports and author responses from that submission.
Round 1
Reviewer 1 Report
Comments and Suggestions for Authors
The paper is nice and on an important topic. Some endoscopic figures would improve the paper, particularly concerning vascular interventions.
THe authors should add a table summarizing the evidence comparing EUS-LB vs PC-LB
I would also comment deeply on the RCTs comparing different needles or sampling techniques for EUS-LB. In this regard , the authors should comment on the state of the art of sempling techiques for EUS-guided tissue sampling (cite the recent multicenter RCT PMID: 35915956 )
Reviewer 2 Report
Comments and Suggestions for Authors
In this manuscript by Upadhye and colleagues titled “Role of Endoscopic Ultrasound in the Diagnosis and Management of Liver Diseases”, the evolving role of endoscopic ultrasound (EUS) in hepatology for both diagnostic and therapeutic applications is discussed. Here are my comments related to this review.
-The abstract section could be improved; for example, authors should be clear about which liver diseases the review is focused on to be diagnosed with endoscopic ultrasound.
-In the introduction section, please include the liver diseases in which endoscopic ultrasound can help diagnose, including its advantages.
-I suggest a Figure in which the authors include an overview of the application of the endoscopic ultrasound and the liver diseases diagnosed by this method, including advantages and adverse events.
-If possible, the authors should include some figures and descriptions using endoscopic ultrasound according to the focus of liver diseases in this review.
-Please define the acronym when they are used for the first time, for instance, HVPG and PBCs.
-There are some words with grammatical errors throughout the manuscript.
Comments on the Quality of English LanguageThere are some words with grammatical errors throughout the manuscript.
Reviewer 3 Report
Comments and Suggestions for Authors
Dear Authors,
This review focuses on the role of endoscopic ultrasound in liver disease diagnosis and management. The work is inappropriate due to a number of significant concerns.
- My primary concern is that similar review articles have been conducted that evaluated the same topic (World J Gastroenterol. 2024 Feb 21;30(7):742–758; World J Hepatol. 2017 Aug 28;9(24):1013–1021; World J Hepatol. 2021 Nov 27;13(11):1459–1483; Gut Liver. 2022 Dec 2;17(2):204–216; World J Gastroenterol. Jun 14, 2024; 30(22): 2920-2922; J Hepatobiliary Pancreat Sci. 2018 Mar;25(3):171-180). There is no clear explanation for the significance of this review. It is necessary to provide a precise explanation of the evidence gap that this review is addressing.
- The lack of methodology used in the selection of the papers given in the review is a crucial problem. Is the authors' selection of publications arbitrary, or does the paper represent the body of extant literature on the topic? Therefore, a search strategy for choosing which papers to include in the review is required.
- Incorporating tables to summarize the findings of all studies is strongly advised.
- Potential limitations of each study included in the review should be sufficiently discussed in the manuscript.
- The journal's referencing guidelines must be adhered to.